# Overview of Risk Factors for Esophageal Squamous Cell Carcinoma in China

**DOI:** 10.3390/cancers15235604

**Published:** 2023-11-27

**Authors:** Erica Conway, Haisheng Wu, Linwei Tian

**Affiliations:** School of Public Health, Li Ka Shing Faculty of Medicine, The University of Hong Kong, 7 Sassoon Road, Hong Kong SAR, China; econway@hku.hk (E.C.); haisheng@connect.hku.hk (H.W.)

**Keywords:** esophageal squamous cell carcinoma, esophageal cancer, cancer etiology

## Abstract

**Simple Summary:**

Esophageal cancer holds the rank of 6th highest mortality rate worldwide and carries a low survival rate of 10–30%. The pathology type of esophageal squamous cell carcinoma (ESCC) represents 90% of esophageal cancers in Asia, Eastern Europe, and Africa. Understanding the etiology of ESCC is key to the formulation of public health prevention and intervention strategies. China, having a high ESCC incidence, has a rich body of literature on the etiology of ESCC but has not seen significant declines in rates in recent decades. Identified were 207 studies that explored various etiologic factors including genetic, gene–environment, dietary, dietary behavior, infection, oral health, environmental, and socioeconomic factors. While further study is essential to better understand these factors’ contributions, public health should prioritize genetic screening for relevant polymorphisms, conduct comprehensive investigations into environmental and dietary influences, consider screening for HPV, enhance oral health education, and consider socioeconomic factors overall to effectively reduce ESCC risk in China.

**Abstract:**

(1) Background: China has the highest esophageal squamous cell carcinoma (ESCC) incidence areas in the world, with some areas of incidence over 100 per 100,000. Despite extensive public health efforts, its etiology is still poorly understood. This study aims to review and summarize past research into potential etiologic factors for ESCC in China. (2) Methods: Relevant observational and intervention studies were systematically extracted from four databases using key terms, reviewed using Rayyan software, and summarized into Excel tables. (3) Results: Among the 207 studies included in this review, 129 studies were focused on genetic etiologic factors, followed by 22 studies focused on dietary-related factors, 19 studies focused on HPV-related factors, and 37 studies focused on other factors. (4) Conclusions: ESCC in China involves a variety of factors including genetic variations, gene–environment interactions, dietary factors like alcohol, tobacco use, pickled vegetables, and salted meat, dietary behavior such as hot food/drink consumption, infections like HPV, poor oral health, gastric atrophy, and socioeconomic factors. Public health measures should prioritize genetic screening for relevant polymorphisms, conduct comprehensive investigations into environmental, dietary, and HPV influences, enhance oral health education, and consider socioeconomic factors overall as integral strategies to reduce ESCC in high-risk areas of China.

## 1. Introduction

The 6th highest mortality from cancer worldwide comes from esophageal cancer, with 544,000 esophageal cancer-related deaths reported in 2020 [1]. Esophageal cancer prognosis is poor; during the period 2010–2014, the age-standardized 5-year net survival was observed to be 10–30% in most countries [2]. In 2018, newly reported esophageal cancers represented 3.2% of all diagnosed malignancies and 5.3% of all cancer-related mortalities, worldwide [1]. There are two primary types of esophageal cancer: squamous cell carcinoma and adenocarcinoma. Adenocarcinoma is recognized to be the dominant type in the Western world [3], whereas squamous cell carcinoma represents 90% of esophageal carcinomas in Asia, Eastern Europe, and Africa [4]. Over 50% of the world’s esophageal cancer cases were identified in China in 2020, with esophageal squamous cell carcinoma (ESCC) as the dominant histological type [5]. China holds the highest ESCC incidence areas in the world, with some areas of incidence over 100 per 100,000 [6]. The survival rate in China has been estimated to have increased from 20.9% to 30.3% in the past ten years; however, most cases were found in late stages, and the prognosis remains quite poor [7]. Despite the extensive public health efforts to investigate ESCC risk factors in China, its etiology is still poorly understood and merits review. To better focus further research, this study aims to systematically review and summarize past research into potential risk factors for ESCC in China.

## 2. Materials and Methods

A systematic literature search was conducted in the databases PubMed, Web of Science, Cochrane, and EMBASE using combinations of the keywords “risk factor”, “risk factors”, “esophageal squamous cell carcinoma”, “oesophageal squamous cell carcinoma”, “China”, and “Chinese” (Table 1). One reviewer used Rayyan [8] to conduct a study title and abstract review, followed by a full-text review. Included were population studies in English, in humans, in populations in China mainland or Hong Kong, and qualitative or quantitative data on ESCC etiology. Exclusion criteria were studies not in English, populations outside of China mainland or Hong Kong, studies focusing on ESCC diagnosis, screening, treatment, or prognosis, studies focusing on dysplasia or precancerous lesions, animal or cell studies, reviews, case reports, studies without qualitative or quantitative risk factor data on ESCC etiology, and irrelevant studies. The study’s PRISMA diagram is shown in Figure 1. Following a full-text review, each study’s main findings were extracted into Excel at the reviewer’s discretion. Data from multivariate analyses were always reported first; if no multivariate data were available, data from univariate analyses were reported. ArcGIS was used to visualize the study population region according to the Chinese province. A meta-analysis was not conducted due to the incomparability of the studies.

## 3. Results

### 3.1. Study Characteristics

From 1890 studies identified, following duplicate study removal, and title, abstract, and full-text review, a total of 202 studies met the inclusion criteria (Figure 1). Through manual searching, five studies were added following a full-text review, resulting in a total of 207 studies being included in this review. An overview of the included study aim, ethnicity, and study count per province is shown in Table 2 and Table 3, and Figure 2, respectively.

### 3.2. Study Type

According to the risk factors identified in the reviewed studies, risk factors were divided into 15 groups including genetics, gene–environment interactions, family history, HPV, *H. pylori*, alcohol and smoking, pickled vegetable consumption, salted meat consumption, tea type consumption, fresh fruit and vegetable consumption, other dietary factors, dietary behavior, oral health, environmental factors, and socioeconomic status.

#### 3.2.1. Genetic

An extensive number of studies report genetic mechanisms related to ESCC risk. Among the 207 studies included in this review, 141 studies investigated genetically related risk of ESCC (Appendix A, Appendix A and Appendix A). SNPs in more than 151 unique genes and their association with ESCC were investigated (Appendix A, Appendix A). Among the genetic variants investigated, eight studies found that various TP53 SNPs increased ESCC risk, albeit with varying statistical significance [9,10,11,12,13,14,15,16]; notably, the p53 rs1042522 Arg/Arg genotype was reported to increase risk by more than five times [9]. Five studies reported significant associations between various SNPs in alcohol metabolism genes ALDH1, ALDH2, or ALDH7 and ESCC risk [17,18,19,20,21]. Notably, Guo et al. reported that among moderate to heavy drinkers, those carrying ADH1B *1/*1 had increased risk by more than 27 times [20]. Four studies showed various SNPs in the XRCC1 DNA repair gene significantly associated with ESCC [19,22,23,24]; a fifth study found no association between the XRCC1 SNP but found an association between SNP haplotype [25]. Additionally, 10 unique miRNA associations were investigated. Other genetically related factors investigated in terms of association with ESCC risk include spectra and frequency of certain gene mutations, short telomere length, gene deletions, loss of heterozygosity, short tandem repeat polymorphisms, and gene–environment interactions (Appendix A, Appendix A and Appendix A).

#### 3.2.2. Gene–Environment Interactions

Twelve studies investigated gene–environment interactions among their primary aims (Appendix A, Appendix A). Notably, Pan et al. found an interaction between short leukocyte telomere length, smoking, and alcohol that increased ESCC risk by almost 17 times [26]. Yang et al. showed that carrying XRCC1 G28152A GA + AA vs. GG and consuming long-term-stored rice increased ESCC risk by more than seven times [24]. Additionally, Tan et al. found an interaction between Thymidylate synthase low-expression genotype 3Rc/3Rc þ 3Rc/2R þ 2R/2R and a low serum folate concentration that increased risk by more than 22 times [27]. Shi et al. found an increased risk of more than eight times among CYP2C19*3 A Allele carriers who drink [28].

#### 3.2.3. Family History

Three studies aimed at investigating family history of ESCC as a risk factor, and a total of about 7% of studies adjusted for family history in their analyses (Appendix A, Appendix A). Chen et al. found an increased risk of ESCC of almost two times among individuals having a first-degree relative with ESCC history, compared to those without a first-degree relative with ESCC history [29]. The same study also reported that having two or more first-degree relatives with a history of ESCC increased ESCC risk by almost three times, and having both parents with a history of ESCC increased ESCC risk by almost eight times [29]. Yang et al. showed a 45% increased risk of ESCC among those with a family history of UGI cancer (including gastric and esophageal cancers) and an additional 20% increase among those who had both parents with a history of UGI cancer [30]. Interestingly, the same study reported that having a spouse with UGI cancer did not increase the risk of ESCC [30]. Nan et al. found that those with a positive family history of ESCC had an earlier age of onset by almost two years compared to those with a negative ESCC family history (51.83 vs. 53.59 years) [31].

#### 3.2.4. HPV

HPV as a risk factor for ESCC has been heavily investigated (Appendix A, Appendix A). Eleven studies across multiple provinces in China measured overall HPV positivity and detected HPV positivity of 0–80% in samples of cases with ESCC [32,33,34,35,36,37,38,39,40,41,42]. Five of these studies also measured HPV-positive controls or healthy distal mucosa samples as comparison groups; no significant trends were found between studies [34,36,38,40,42]. Shuyama et al. compared HPV prevalence in high versus low-risk ESCC risk areas and they detected HPV in 65% and 6% of cases in high and low-risk areas, respectively [41]. Multiple studies reported on high-risk HPV prevalence in ESCC cases, with emphasis on HPV-16 and/or 18. Five studies assessed the association between HPV (including studies that looked at HPV overall, HPV high-risk types, or HPV serum antibodies) and ESCC [34,36,42,43,44]. Among the studies that looked at HPV overall, two studies reported an increased risk of ESCC of 1.56 and 6.4 times [34,42]; one study found no significant association among non-smokers or drinkers but showed an increased risk of two times among smokers and an increased risk of more than ten times among individuals who smoke and drank [36]. Two studies from neighboring provinces studied HPV-16 and the risk associated with ESCC; Guo et al. reported an increased risk of ESCC by nearly thirteen times; Kamangar et al. found no significant association [34,44]. Lastly, Yang et al. investigated the interaction between p53 polymorphisms and HPV-16 seropositivity. They found that among individuals carrying p53 rs1042522 Arg/Arg or Arg/Pro and HPV-16 seropositivity, ESCC risk increased more than nine times, compared to individuals HPV-16 seronegative and carrying the Pro/Pro genotype, and that risk increased to 27 and 13 times among smokers and drinkers, respectively [9].

#### 3.2.5. *Helicobacter pylori*

Three studies investigated *Helicobacter pylori* (*H. pylori*) as a potential risk factor (Appendix A, Appendix A). Li et al. investigated *H. pylori* infection rate in ESCC samples from the three high-risk regions of Linzhou, Shantou, and Shaanxi and reported no differences, but noted that the infection rate was very high overall (estimated ≥70%) [45]. They used PCR to investigate *H. pylori* 16S rRNA in ESCC and healthy tissues and found 63% and 27% of *H. pylori* among cases and controls, respectively. Xue et al. investigated the association between *H. pylori* and ESCC in Hebei province and found no significant association; however, the ESCC sample size was small [46]. In contrast, Wang et al. conducted a multivariate analysis in Jiangsu and showed that *H. pylori* infection increases ESCC by more than three times [47]. Gastric atrophy, which in most cases is caused by chronic *H. pylori*, was reported as an etiological factor in three studies. A recent study in 2020 reported a 61% increased risk of ESCC associated with gastric atrophy, signified by having a serum pepsinogen level I of <55 µg/L [48]. Similarly, Ren et al. reported a 56% increased risk of ESCC associated with gastric atrophy, defined as PG I/II ratio ≤ 4 ng/mL; however, quartile models and continuous models reported no association [49]. In contrast, Xue et al. looked at PG I ≤ 70 ng/mL, PG I/II ratio ≤ 3, or 70 ng/mL and PG I/II ratios and ESCC and found no significant associations [46]. In terms of interactions, as mentioned earlier, Ekheden et al. reported an additive interaction between poor oral health and gastric atrophy that increased ESCC risk by 28%, although this finding was not statistically significant [48].

#### 3.2.6. Alcohol and Smoking

Multiple studies reported associations between alcohol consumption and ESCC (Appendix A, Appendix A). Xu et al. reported that alcohol increased ESCC risk almost 1.7 times among men and women [50]. Two studies reported that alcohol consumption increased ESCC risk by more than two times among men [24,51]. Guo et al. found that alcohol consumption increased ESCC risk by more than 3 times among heavy drinkers, and by more than 1.5 times among those who consumed alcohol for more than 30 drink-years [20]. Similar to alcohol, multiple studies reported smoking increased ESCC risk by 1.5–2.6 times [50,52,53,54]. Chen et al. found that smoking tobacco for 30 or more years increased ESCC risk by more than 4.5 times [55]. Kumagai et al. reported that smoking 20 or fewer cigarettes a day for more than 20 years was associated with an increased risk of ESCC by more than 1.5 times; however, smoking more than 20 cigarettes a day for 20 years or less, or smoking 20 cigarettes a day for more than 20 years, did not find a significant association, so there may be an issue in this study [56]. The interaction of smoking and alcohol was also reported to affect ESCC risk. Wu et al. found that the joint effect of 40 or more cigarettes per day and 500 mL of alcohol/week increased ESCC risk by more than seven times, compared to those who never smoked or drank [52]. Chen et al. reported that consuming alcohol and smoking tobacco interaction increases the risk of ESCC by almost 44 times [55]. When factoring in duration and dose intensity of alcohol consumption, the same study found that consuming alcohol for 20 years or more, and consuming 60 or more grams, was associated with an astounding increase of 183 times in ESCC risk.

#### 3.2.7. Pickled Vegetable Consumption

Among the studies included in this review, three studies found significant independent associations between pickled vegetable consumption and increased ESCC risk, and one study reported no association (Appendix A, Appendix A). Yu et al. compared the lifestyle of immigrants from the high-risk area of Henan who immigrated to Caihu, Hubei, a lower-risk region, and host residents of Hubei, and found that immigrant residents had a higher consumption of pickled vegetables than host residents [57]. Additionally, Peng et al. conducted a study in Fujian province, also a high-risk region, and showed a significant positive association between consumption of pickled vegetables and ESCC among ESCC cases and healthy controls, as well as an additive interaction between CYP219 polymorphism and pickled vegetable consumption [58]. Wang et al. reported that in their study conducted in Huai’an, Hebei Province, also a high-risk region, a univariate significant association was found between consumption of pickled vegetables and ESCC; however, this association was not present in the multivariate analysis [47]. In contrast, in Tran et al.’s study looking prospectively at the Linxian Intervention Trial, among which almost 2000 individuals developed esophageal cancer during the follow-up period, they reported no significant associations between pickled vegetable consumption and ESCC [59].

#### 3.2.8. Salted Meat Consumption

Two studies reported on salted meat consumption (Appendix A, Appendix A). In terms of salted meat consumption amount, Lin et al. reported a slightly increased risk of 18% associated with consuming 50 g/week of salted meat in Sichuan province [60]. The same study reported that the interaction of high salted meat intake, defined by more than 90.8 g/week, with alcohol and smoking, increased ESCC risk by more than 12 times. In terms of salted meat intake frequency, the same study reported high salted meat intake, defined as intake of four or more times per week, was associated with a seven-times increased ESCC risk. Zhao et al., who also investigated high salted meat consumption frequency in Sichuan province, but defined high salted meat consumption as consuming salted meat one or more times per week, reported that this frequency of salted meat consumption increased the risk to ESCC by more than two times [61].

#### 3.2.9. Tea Type Consumption

Four studies reported on green tea consumption among ESCC cases and controls (Appendix A, Appendix A). Two studies reported green tea as a protective effect against ESCC. Wang, Z. et al. reported an 87% decreased risk of ESCC associated with green tea drinking [47], and Wang, J.M. et al. reported a 74% decreased risk of ESCC associated with green tea drinking in women [51]. In contrast, Yang et al. reported green tea drinking as increasing the risk to ESCC by more than 1.5 times among men [62], and Lin et al. found no significant association between green tea drinking alone and ESCC [63]. The same study showed that black tea, on the other hand, increased the risk of ESCC by almost two times among individuals who consumed >300 g/month of black tea [63]. One study investigated jasmine tea consumption and reported no significant association between jasmine tea consumption among males and females combined; however, they did report that high consumption of jasmine tea among males was associated with a 1.68 increased ESCC risk [64].

#### 3.2.10. Fresh Fruit and Vegetable Consumption

Four studies investigated associations between fresh fruit and vegetable intake and ESCC risk (Appendix A, Appendix A and Appendix A). Gao et al. reported ESCC risk reductions of 47% and 54% associated with the consumption of 189 times or more of fresh fruit consumption, and 1230 times or more of fresh vegetable consumption per year, respectively [65]. Peng et al. reported that consuming fresh fruit and vegetables of 400 g or more per day decreased ESCC risk by 80%, although this was not statistically significant [58]. Additionally, Liu et al. reported having a dietary food pattern containing higher pattern weight from vegetables and fruits was associated with a 30% reduction in ESCC risk [66]. Tran et al. found a reduced risk of 20% associated with fresh fruit intake of more than 13 times per year but did not find any significant association with fresh vegetable consumption in any amount of intake [59].

#### 3.2.11. Other Dietary Factors

Other factors related to dietary consumption were also widely investigated (Appendix A, Appendix A and Appendix A). Significant inverse associations reported between nutritional factors and ESCC risk include dietary selenium [15], dietary n-3 long-chain polyunsaturated fatty acid pattern consumption [67], dietary monounsaturated and polyunsaturated fatty acids [68], high vitamin D-3 and beta-carotene levels [69], higher intake of certain flavonoids (isoflavones, daidzein, genistein, and glycitein, specifically), moderate anthocyanidin consumption [70], as well as “prudent” food patterns (higher tofu and bean-curd, dry beans, seeds, wheat, rice, etc.) [66], and peanut consumption of one or more times per week (also a dose–response relationship) [71]. Significant positive associations reported between nutritional factors and ESCC risk include dietary even-chain unsaturated fatty acid pattern consumption [67], dietary inflammatory nutrients [72], consumption of processed food or alcohol drinking patterns [66], high mycotoxin exposure [73], consumption of pork braised in brown sauce, old stocked rice, high amounts of chili (among men) [51], or milk or dairy products [65]. One study assessed the daily consumption of scalding hot food, fried food, moldy food, or pickled vegetables, compared to a diet excluding those factors, and found an associated increased risk of ESCC of four times [74]. One study found moldy food consumption increased ESCC risk [65], whereas Tran et al. found consumption of moldy bread or moldy vegetables had no significant association with ESCC [59].

#### 3.2.12. Dietary Behavior

In this review, five studies reported hot food or drink consumption was associated with a more than two-times increased risk of ESCC (Appendix A, Appendix A and Appendix A) [51,58,65,75,76]. Among these studies, Peng et al. also found an additive interaction between CYP2C19 GA/AA genotype polymorphism and hot beverage and food consumption [58]. Additionally, three studies reported that drinking very hot tea was associated with a more than 1.5-times increased risk (odds ratios ranging from 1.67 to 2.48) [62,63,76]. Among these eight studies, Tai et al. specifically investigated the consumption of high-temperature versus low-temperature water and found an increased risk of ESCC of more than two times [76]. Also among these studies, Zhao et al. reported that waiting 10 or more minutes between water boiling and drinking decreased cancer risk by more than 80% [75]. One study by Tran et al. found no association with the consumption of hot liquids [59]. The behavior associated with alcohol consumption has also been previously investigated. Sun et al. found that drinking alcohol before a meal at mealtime, compared to drinking alcohol and eating at the same time, increased ESCC risk by 1.76 times and that drinking alcohol outside of mealtime compared to drinking alcohol at mealtime increased the ESCC risk by more than six times (Appendix A, Appendix A) [77]. Additionally, they showed that drinking before mealtime increased the risk by more than 3.5 times among heavy drinkers [77]. Additional dietary behavior investigated includes fast eating, as well as the period between food consumption and bedtime. Wang et al. reported fast eating increased the risk of ESCC by more than three times (Appendix A, Appendix A) [47]. Similarly, Zhao et al. additionally reported that a more than 50% decrease in ESCC is associated with taking 15 min or more to finish a meal (Appendix A, Appendix A) [75].

#### 3.2.13. Oral Health

Several studies reported an associated risk of ESCC with poor oral health (Appendix A, Appendix A). Abnet et al. reported a relative risk of 1.3 associated with tooth loss and ESCC in Linzhou, Henan Province [59]. Similarly, Chen et al. reported a nearly 1.5-times increase in the risk of ESCC, associated with the loss of six or more teeth after the age of 20, compared to losing no teeth after the age of 20, but in Jiangsu Province [78]. The same study showed that brushing teeth less than or equal to one time per day was associated with a 1.8-times increased risk of ESCC, compared to brushing teeth two or more times per day. In another study conducted in Jiangsu Province, Chen et al. reported finding individuals with ESCC had less diversity of oral bacteria compared to healthy individuals [79]. Zhao et al. reported an association of ESCC with poor oral hygiene in combination with genetic interactions in Jiangsu Province [80]. They reported a nearly five-times increase in risk of ESCC from a synergistic interaction between the PLCE1 gene SNP rs3765524 TT and tooth brushing less than two times per day [80]. In addition, they reported a two-times increased ESCC risk of an interaction between ADH1B rs1159918TT and tooth loss. One study by Ekheden et al., also in Jiangsu Province, reported additive interactions between gastric atrophy and poor oral health. They reported a nearly 1.3-times increased risk of gastric atrophy combined with once-a-day or no tooth brushing; however, this result was not statistically significant [48].

#### 3.2.14. Environmental Factors

Three studies reported etiological factors related to environmental exposures (Appendix A, Appendix A and Appendix A). In two studies, Lian et al. found phytoliths resembling prickle hair from wheat bract in ESCC tumor tissue samples [81,82]. Additionally, Yu et al. reported drinking river water increased risk by more than four times in men and more than eight times in women [24]. Lastly, Zhao et al. showed an increased risk of ESCC among urinary exposures to N-nitrosamines [83].

#### 3.2.15. Socioeconomic Status

Other etiological factors of ESCC identified in multiple studies in this review include socioeconomic factors (Appendix A, Appendix A and Appendix A). Three studies looked at education level as a risk factor. Tran et al. found having 1–5 years of education, having completed primary school, or having completed middle school reduced ESCC risk by 13%, 22%, and 43%, respectively [59]. Gao, P. reported having completed primary school, primary high school, or secondary high school or more was associated with a 26%, 40%, and 40% reduction in ESCC risk, respectively [84]. Gao, Y. et al. found that having 6–9 years of education was associated with a 25% reduction in ESCC risk [65]. Gao, Y. et al. also found that having a higher education (secondary high school or above) was associated with a reduced risk of ESCC by 40%, but this association was not statistically significant. Household size as a socioeconomic variable was investigated in two studies. Y. Gao et al. found that having four or more people in each household was associated with a 29% increased risk [65]. Gao, P. et al. found that having six or more people in the household increased risk by 63% [84]. In addition, performing high amounts of physical labor was shown to increase ESCC risk [84]. Having water piped in the home [59], having a larger house area per person [84], having a higher wealth score [84], and owning certain appliances including owning a refrigerator for >5 [84] or ≥8 years [65], were included among the socioeconomic factors found to reduce ESCC risk.

## 4. Discussion

This review highlights the extensive amount of research that has been conducted to investigate multiple mechanisms related to ESCC etiology in China. The most investigated risk factors were genetically related, followed by diet and dietary habits, HPV, gene–environment interactions, oral health, family history, *H. pylori*, and socioeconomic factors.

Almost 70% (68.11%) of included studies investigated genetically related etiologies of ESCC, among which most investigated genetic polymorphisms. The most investigated polymorphism was that on the TP53 gene. The TP53 gene encodes p53, which primarily functions as a tumor suppressor protein, which plays a critical role in cancer development [85,86]. TP53 mutation is the most occurring mutation found in human cancer, with over 50% of cancers holding mutations in the TP53 gene [87]. TP53 mutation leads to p53 loss of functions that are needed to prevent tumor growth, which can lead to cell multiplication and cancer development [88,89]. Because of TP53′s role in cancer development, much research has been conducted looking at TP53 mutations and associations with ESCC. In addition to TP53 polymorphisms, the genetic variations included in this review include SNPs of over 151 unique genes, gene deletions, loss of heterozygosity, missense point mutations, and other genetic variations. Based on the extensive evidence of genetically related risk of ESCC shown in Appendix A, Appendix A, it is clear that there is a genetic influence on ESCC risk. However, due to the heterogeneity of studies, it was challenging to compare the influence of many different genetic variations measured across studies in a variety of regions, and a meta-analysis was not performed. A meta-analysis may be merited, however, in the future, if more evidence of the same genetic variations becomes available in comparable populations.

The significant portion of studies identified in this review that focused on genetically related etiologies underscores the importance of genetic susceptibility in advising public health prevention and intervention measures. These measures include genetic screening, early detection, personalized prevention strategies, targeted intervention, and public education and awareness. Genetic screening of polymorphisms in high-risk populations could identify genetic mutations associated with increased ESCC risk, could encourage individuals at high risk to be more proactive about endoscopic screening for early cancer detection when treatment outcomes are more favorable, and could provide eligibility information for potential gene therapy interventions. Because most cases are identified in late stages [90], more emphasis on early screening, both genetic and endoscopic, prevention, and public education and awareness is imperative. Guidelines for practitioners for advising individuals carrying high-risk mutations, e.g., TP53 mutations, should be developed and readily/widely available. Guidelines might include younger, more frequent, and more aggressive cancer screening for individuals at higher risk. Policies should be implemented to fund genetic screening in high-risk populations. This review highlights the need for genetic susceptibility screening and the potential for more targeted intervention based on genetic risk factors of ESCC to be prioritized, especially in Chinese communities with significantly higher mortality rates of ESCC such as in Linzhou. Although genetic susceptibility was the most frequently researched etiologic factor identified in this review, its influence relative to and interaction with other factors is still unknown. Therefore, a comprehensive approach that measures genetic as well as other factors is crucial for effective public health ESCC intervention and prevention.

In addition to the extensive investigation of genetically related risk factors, this review also delved into other critical aspects contributing to ESCC risk, encompassing gene–environment interactions, family history, diet, dietary behavior, environmentally related risk factors, *H. pylori*, HPV, oral health, and socioeconomic factors. Many of these studies relied on retrospective qualitative data gathered through interviews, which carries the potential for recall bias. Despite these limitations, many studies identified significant associations.

The twelve studies that focused on gene–environment interactions highlighted in this review (Appendix A, Appendix A), particularly the studies that found associations that increased ESCC risk by almost 17 [26] and more than 22 [27] times, emphasize the significance of gene–environment interactions in ESCC development. Genetic polymorphisms or genetic changes can affect one’s susceptibility to environmental exposures and vice versa. For example, the increased risk from the positive multiplicative interaction between an XRCC1 polymorphism and consuming long-term-stored rice, found by Yang et al., may suggest that the dietary habit of consuming long-term-stored rice may worsen the effects of an XRCC1 polymorphism affecting DNA damage repair. This may lead to a synergistic effect on the development of ESCC, meaning that the combined effects of both factors are greater than the sum of each of their individual effects. Another example is the cumulative interaction between short leukocyte telomere length, smoking, and alcohol consumption found by Pan et al. This finding may suggest that smoking and alcohol consumption could cause oxidative stress, which would further impact individuals with shorter telomeres, further exacerbating the risk of all individual effects when combined. Additionally, the interaction between the Thymidylate synthase low-expression genotype and low serum folate concentration reported by Tan et al. [27] may suggest that consuming low amounts of folate may worsen DNA damage from the Thymidylate synthase polymorphism, leading to increased ESCC risk. This presence of a complex interplay between environmental exposures and genetic susceptibility in ESCC development suggests that a comprehensive approach to cancer prevention that includes genetic susceptibility, gene–environment interactions, and other etiologic factors is needed for effective ESCC mitigation. By incorporating information about gene–environment interactions into public health education programs, public health authorities can raise awareness about the complexity of cancer development and encourage individuals to be more proactive about reducing their risk of developing ESCC.

Three studies in this review reported family history as a potential ESCC etiologic factor, but because of the complexity of family history, concluding family history alone is challenging. Family history as an etiology of any disease can mean one’s risk might be affected by inherited genetic variations, similar living environments or lifestyles as family members, or both, and therefore contains many different factors. Only 7% of the studies included in this review adjusted for family history in their study; the lack of higher adjustments or considerations for family history makes sense as family history can mean a range of influencing factors, which makes it challenging to adjust for. Additionally, because most studies do not have exposure information of relatives, it is challenging to distinguish the influence of environmental factors from genetic influence. Instead of family history as an etiologic factor, considering genetic, environmental, and the interaction between them may provide more insight into ESCC development.

Multiple studies identified alcohol and tobacco as potential risk factors in Jiangsu, Shandong, Hebei, and Gansu, among other provinces, and many studies acknowledged alcohol and tobacco consumption as well-established risk factors and adjusted for them in their analyses. Despite this evidence, a study conducted in Linzhou (formally known as Linxian), in 1989, which at the time had ten times higher rates of esophageal cancer mortality than China’s average, and still has one of the highest ESCC mortality rates in the world today, reported close to zero tobacco and alcohol consumption among females, little alcohol consumption in males, and only slightly higher tobacco usage among male ESCC cases compared to controls [91] (the study was not included in this review because it did not report ESCC-specific data). This review identified multiple studies that found alcohol and tobacco, and interactions with them, as contributing to ESCC development. However, this may vary by region, lifestyle, and culture. Further research including community dietary studies and risk factor comparisons by region may help address the role of alcohol and tobacco relative to other potential etiologies. Public health efforts should prioritize recognizing esophageal cancer as a heightened risk associated with tobacco and alcohol use, especially in high-incidence ESCC regions. Promoting greater public awareness of the link between alcohol and tobacco use and esophageal cancer is advisable, as well as the implementation of stricter tobacco and alcohol control measures, alongside the provision of well-funded, accessible programs for tobacco and alcohol use reduction.

Frequently consumed in high-risk areas of China, pickled vegetables and salted meat were reported by multiple studies in this review to be associated with ESCC etiology. The pickling of vegetables via fermentation and curing meats via the addition of nitrite or nitrate salts can produce N-nitroso compounds [60] or mycotoxins, [58] both considered carcinogens [92,93,94]. It is also possible that consuming high amounts of salt could lead to esophageal tissue injury, which could increase esophageal cancer risk through repeated esophageal injury, and that consuming high amounts of pickled vegetables could contribute to gastric atrophy, another risk factor identified in this review, through mucosa inflammation from repeated high-acid pickled vegetable consumption. Not all studies identified in this review identified pickled vegetables and salted meat as risk factors. As with many dietary studies, this inconsistency between studies could stem from dietary questionnaire recall bias, poor questionnaire design, or little reported variation between individuals’ dietary habits. The evidence suggests further exploration into diet culture in high-risk areas is imperative, particularly by comparing them with drastically lower ESCC incidence, such as the contrast between the high-incidence areas of Linzhou and adjacent lower-incidence areas. Furthermore, considering the possibility of contamination of various origins such as water, air, soil, and sharp micro contaminants, e.g., silica or microplastics, adds further complexity to dietary-related risk factors. Accordingly, environmental health assessments should be conducted to further assess food contamination and safety. Through exploring dietary culture as well as environmental health factors, public health authorities can implement tailored interventions that promote healthier dietary choices to reduce ESCC risk. These interventions might include educational campaigns on higher-risk foods and policies to reduce consumption of pickled and salted foods. Such holistic efforts hold the potential to improve food safety and reduce ESCC risk.

The consumption of hot food or drink was identified in eight studies as a potential ESCC etiologic factor. Very hot beverages are designated as Group 2A carcinogens for ESCC by the IARC monographs, although they mention that the evidence in humans is limited [93]. A few potential mechanisms of hot food or drink increasing ESCC risk are through physical esophageal damage leading to chronic inflammation of the esophageal mucosa, or the reduced ability of the mucosa to prevent exposure to carcinogenic substances, and the carcinogens themselves increasing risk to ESCC. Also identified in this review were multiple studies investigating tea consumption as a potential etiologic factor, with a range of outcomes. Among the studies that identified tea consumption as a risk factor, a potential mechanism could be a thermal injury. Other potential mechanisms include exposure to contaminants (e.g., mold) from fermented tea (black tea) old tea (some tea is intentionally consumed after prolonged storage) or water contamination. Current evidence suggests a further review of hot food and beverage consumption, as well as tea consumption, as risk factors of ESCC, as well as a comprehensive exam of potential mechanisms involved, such as thermal injury, change in salivary gland action, or exposure to contaminants. Additionally, it is critical to further develop targeted public health interventions, such as reducing high-temperature food and drink consumption.

HPV has been heavily investigated as a potential ESCC etiologic factor because of the extensive evidence of HPV’s association with cervical cancer [95,96]. In terms of ESCC etiology, it has been proposed that some proteins made by HPV, especially E6, are oncoproteins that can inactivate host proteins, namely p53, and may cause mutations in host cell DNA and lead to ESCC [97]. Across eleven studies in this review, there was a wide range of HPV-positive prevalence rates reported among various provinces in China. Among the studies that looked at the association between HPV and ESCC risk, some studies found increased risk to ESCC, and others did not. This mixed level of detection, and lack of trends among associations, may point to a true lack of HPV prevalence in ESCC tissue and its limited evidence as an etiologic factor, but could also be due to the varying methods in sample processing or detection (e.g., PCR-based assays or serological tests), varying viral loads in ESCC samples and controls, regional differences, and other confounding factors that include alcohol or tobacco use. The presence of HPV in ESCC tissues, and the significant associations found in some studies, though limited, underscores HPV as a potential ESCC etiologic factor. More research is needed to further understand HPV’s impact on ESCC risk and its mechanisms, as well as to inform early ESCC detection efforts, e.g., HPV vaccination strategies, and any related regional disparities.

Additional environmental factors were explored in four studies. Two of the studies identified siliceous prickles resembling prickle hair from wheat bracts, in ESCC tissues [81,82]. Sharp micro-silica could potentially physically damage esophageal tissue and even provide a physical attachment point for cancer cell proliferation. Silica fibers could come from prickle hairs in wheat flour or wheat products as proposed in the two mentioned studies or could come from other sources entirely as silica is found in many foods and naturally in the environment. Additional environmentally related evidence identified in this review includes river water consumption as a risk factor [24], and urinary exposure to N-nitrosamines [83]. The available research on environmental factors associated with ESCC in China is limited but suggests that further investigation is warranted due to the potential impact of environmental exposures on the development of ESCC.

Several studies in this review reported that poor oral health, such as tooth loss, teeth brushing frequency, and oral bacteria diversity, may increase ESCC risk. Poor oral health could also play a role in ESCC development by increasing one’s risk for gastric atrophy, changing oral bacteria that affect how orally exposed carcinogens are digested, and increasing local and systemic body inflammation, leading to increased ESCC risk. These studies, four of which were conducted in Jiangsu, used questionnaires as some or part of their method of assessing oral health, which comes with recall bias. Additionally, the identified associations could also be confounded by diet or socioeconomic hardship (e.g., lack of access to dental care, or overall healthcare access as a whole), amongst other factors. For example, Abnet et al.’s study [98] reported an increased risk of ESCC associated with tooth loss but did not address the potential reasons behind the tooth loss, which makes it hard to differentiate whether tooth loss itself increased ESCC risk, or if other factors increased ESCC risk and also contributed to tooth loss. Given these limitations, studies identified in this review suggest there is a link between poor oral health and ESCC. Public health education around improving oral health should be prioritized in high-risk communities to reduce ESCC risk.

Socioeconomic status (SES) factors, such as education level, household size, or appliance ownership, were also highlighted in multiple studies as potentially affecting ESCC risk. This makes sense, as SES indicators together may contribute to disease risk. For example, individuals who have received more formal education may have higher income opportunities, which may come with access to more frequent medical care, and earlier cancer detection. Individuals with more formal education may also have selectively higher access to endoscopic screening for ESCC and may therefore more frequently get screened. Many of the diet and lifestyle factors may be influenced by SES. As addressed in multiple studies, the importance of SES factors in disease development, including ESCC, is significant. Further efforts should be made to prioritize research into finding more SES factors that may influence ESCC development and allocating resources to mitigate those.

This review has several limitations. Only studies published in English were included in this review, which could underestimate or inaccurately represent the past or current research in China. Additionally, search terms used for the literature search were chosen to broadly encompass most research in the field; however, the search terms chosen could have also limited the studies included. Some studies included in this review did not account for the cancer history of all their study participants, or the history of cancer in study participants’ families, which could confound some of the study outcomes. Furthermore, the reported findings from each reviewed study do not include all results reported by each study, but instead include the most relevant and statistically significant results (*p* < 0.05) as decided by the reviewer; therefore, there are additional factors reported in these studies that may contribute to or be associated with ESCC risk. Also, this study had a single reviewer, introducing possible bias that could have been mitigated with a team of reviewers. Lastly, as this was a qualitative review of studies based on inclusion and exclusion criteria, this comes with some inherent selection bias of studies to some extent.

## 5. Conclusions

In summary, this review underscores the variety of factors that may contribute to ESCC development in China, including genetic variations, gene–environment interactions, dietary factors such as alcohol and tobacco use, pickled vegetables and salted meat, hot food or drink consumption, infections such as HPV, poor oral health, gastric atrophy, environmentally related factors, and socioeconomic factors. While further study is essential to better understand the interactions and contributions of these potential etiologic factors, public health measures should prioritize genetic screening for relevant polymorphisms, comprehensive investigations into environmental and dietary influences, enhanced oral health education, implement enhanced HPV screening in high-risk ESCC areas, and pursue initiatives aimed at improving socioeconomic status overall. These comprehensive measures are critical to effectively reduce ESCC risk and improve public health outcomes in high-risk ESCC areas of China.

## Figures and Tables

**Figure 1 cancers-15-05604-f001:**
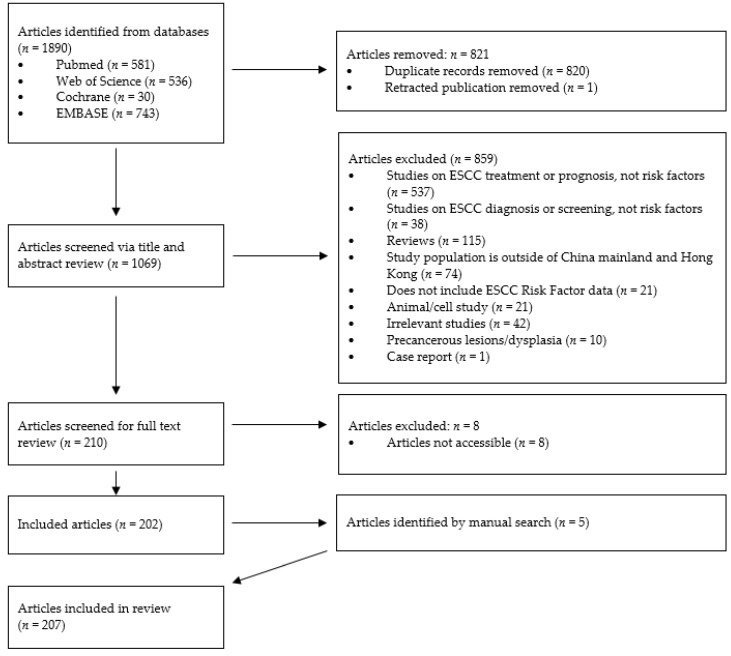
Study PRISMA diagram.

**Figure 2 cancers-15-05604-f002:**
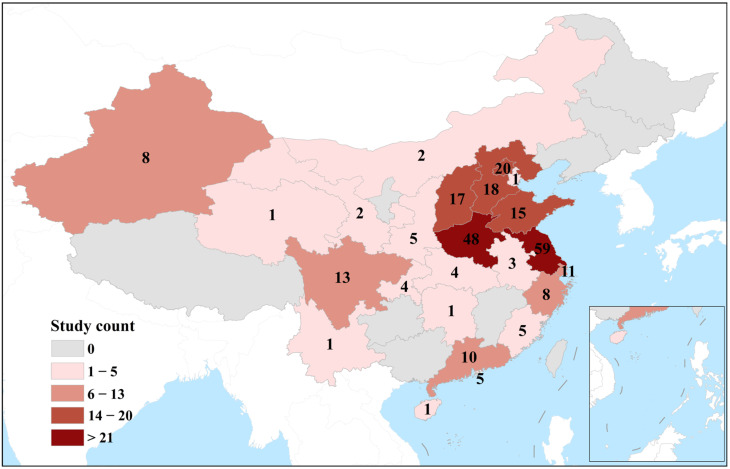
Estimated study count per Chinese province. Many studies included populations from two or more provinces; four studies did not report from which Chinese province data were sourced from, so these were not counted.

**Table 1 cancers-15-05604-t001:** Database search terms used for each online database.

Database	Search Terms
Pubmed	(“risk factor” OR “risk factors”) AND (“esophageal squamous cell carcinoma” OR “oesophageal squamous cell carcinoma” OR “esophageal squamous-cell carcinoma” OR “oesophageal squamous-cell carcinoma”) AND (“China” or “Chinese”) Filters: Human, English
Web of Science	(“risk factor” OR “risk factors”) AND (“esophageal squamous cell carcinoma” OR “oesophageal squamous cell carcinoma” OR “esophageal squamous-cell carcinoma” OR “oesophageal squamous-cell carcinoma”) AND (“China” or “Chinese”)
Cochrane trials	All text: (risk factor OR risk factors) AND (China OR Chinese) AND (esophageal squamous cell carcinoma OR oesophageal squamous cell carcinoma OR oesophageal squamous-cell carcinoma OR esophageal squamous-cell carcinoma) Filters: English, Human
Embase	All text: (risk factor OR risk factors) AND (China OR Chinese) AND (esophageal squamous cell carcinoma OR oesophageal squamous cell carcinoma OR oesophageal squamous-cell carcinoma OR esophageal squamous-cell carcinoma) Filters: English, Human, has an abstract, and exported only articles. Selected exclusions: conference abstracts, reviews, articles in press, preprints

**Table 2 cancers-15-05604-t002:** Studies divided into groups based on potential intervening risk factors.

Study Group	Count (*n* = 207)
Genetic	129 (62.3%)
Diet/dietary habits	22 (10.6%)
HPV	19 (9.2%)
Other	11 (5.3%)
Gene–environ. interaction	10 * (4.8%)
Multiple risk factors	6 (2.9%)
Oral Health	5 (2.4%)
Family history	3 (1.4%)
*H. pylori*	2 ** (1.0%)

* Two additional studies investigated risk associated with gene–environment interactions but were counted under the Oral Health category in this list. ** 1 additional study investigated risk associated with *H. pylori* but was counted under the Multiple risk factors category in this list.

**Table 3 cancers-15-05604-t003:** Study population ethnicity.

Study Population Ethnicity	Count (*n* = 207)
Not reported	138 (66.7%)
Han	66 (31.9%)
Kazakh	2 (1%)
Uyghur, Kazakh, or Han	1 (0.5%)

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
