# Peer review of "Overview of Risk Factors for Esophageal Squamous Cell Carcinoma in China"

_cancers, 2023, doi:10.3390/cancers15235604_

Round 1

Reviewer 1 Report

Comments and Suggestions for Authors

   This is a systematic review focused on the risk factors for esophageal squamous cell carcinoma in China. The purpose, methods, results and assessments are excellent. English sentences are so fluently. Followings are my comments for revision.

Points for consideration,

1.     Authors frequently uses ‘compared to’, but there is a difference between ‘compare to’ and ‘compare with’. I think most cases may better be used with ‘compare with’.

2.     ‘H. pylori’ should be typed in italic stile.

3.     Line 171: H. pylori is used with Helicobacter pylori. The first appearance of the words should be accompanied with the full description with abbreviation.

4.     Line 186: I cannot understand the meaning of ‘or 70 ng/ml and PG I/II ratios’,

5.     Line 206: ‘500m of alcohol’ is not adequate’ because the concentration of alcohol is various.

6.     Line 313: ‘due to’ may better be ‘with’.

7.     Line 323: Is ‘five time an increase’ OK?

8.     I want to see a list table of the presented risk factors, so that the readers could understand the factors at a glance, if possible. In that table, positive and negative risk factors are separately grouped, I wish.

Comments on the Quality of English Language

Excellent.

Author Response

Dear Editor,

Thank you for reviewing our manuscript.

Regarding your suggestions, thank you for your attention to detail. I have made some adjustments according to your points, e.g. H. pylori is now in italics.

Regarding your comment number 8, I have adjusted Table 2 accordingly. I hope that this will improve the readability of the paper. I chose to emphasize the presented work in aggregate,  instead of classifying risk factors as positive or negative, because of the limited comparability of the various studies. I hope that readers will refer to the Supplementary Tables S1-S10 if they have specific questions about the risk factors investigated and the associated results of each study.

Respectfully,

Erica Conway

Reviewer 2 Report

Comments and Suggestions for Authors

This paper provides an overview of risk factors for esophageal squamous cell carcinoma (ESCC) in China, a country with a high incidence of ESCC, by reviewing a large number of articles. Although only data from China are included in this paper, it is considered to be a meaningful paper considering the high incidence of ESCC in China.

1.     The risk factors for ESCC in China are described in text, but it would be easier for readers to understand if there are Tables that lists particularly important and intervening risk factors.

2.     Please check the following in several places.

Are the numbers of papers in Figure 1 correct? For example, the authors picked up a total of 1890 papers from several databases and excluded 823 of them, so there would be 1067 papers remaining, but Figure 1 lists 1069. There seem to be other errors besides this in Figure 1.

There are errors in the correspondence between the text and the supplementary tables. In particular, there are many errors in the text corresponding to Tables S5-S7.

Author Response

Dear Editor,

Thank you for reviewing our manuscript and for your suggestions.

Regarding your comment 1:

You make a good point. I have adjusted Table 2 to reflect my intentions to express that these risk factors may intervene. I hope that this will improve the readability of the paper. I chose to emphasize the presented work in aggregate, instead of by the intervening risk factors, because of the little comparability among the studies.

Regarding the first part of your comment 2:

Thank you for your attention to detail. In the initial round of exclusions, 821 papers were identified as duplicates (820) or retracted publications (1). Subsequently, 1069 papers were screened via title and abstract review. Upon revisiting the data, I noted that the counts of duplicates were not accurately reflected in Figure 1. I have corrected Figure 1 to reflect this in the manuscript.

Furthermore, I’ve updated the counts for the title and abstract review exclusions to 859 by consulting Rayyan, the program used for article review, which saves the counts of each article exclusion category. It appears that the final counts that Rayyan recorded were not accurately transferred to Figure 1, so I have updated that accordingly. Please note that the overall initial and final paper counts remain consistent, and that no analyses or decisions were altered during this correction process.

Finally, I have included, as supplementary materials, the bibliographies of both the initial set of 1890 studies and the final inclusion of 207 studies for reference and transparency.

Regarding the second part of your comment 2:

Thank you for pointing that out. I have checked the mentioned text that refers readers to corresponding tables. I have updated that text to match that of the supplementary tables in areas that were overlooked.

Respectfully,

Erica Conway

Round 2

Reviewer 2 Report

Comments and Suggestions for Authors

Thank you for giving me the opportunity to peer review. I confirm that suitable alterations and corrections have been made to the points I raised.